# The Emerging Role of CKAP4 in GI Cancer: From Molecular Pathways to Clinical Applications

**DOI:** 10.3390/curroncol32100561

**Published:** 2025-10-07

**Authors:** Markos Despotidis, Orestis Lyros, Tatiana S. Driva, Panagiotis Sakarellos, René Thieme, Andreas Mamilos, Stratigoula Sakellariou, Dimitrios Schizas

**Affiliations:** 1First Department of Surgery, National and Kapodistrian Laikon General Hospital, University of Athens, 10679 Athens, Greece; psakarell@med.uoa.gr (P.S.); dschizas@med.uoa.gr (D.S.); 2Fourth Department of Surgery, Attikon University Hospital, National and Kapodistrian University of Athens, 10679 Athens, Greece; 3First Department of Pathology, Medical School, National and Kapodistrian University of Athens, 10679 Athens, Greece; tatianadriva@gmail.com (T.S.D.); sakelstrat@med.uoa.gr (S.S.); 4Department of Visceral, Transplant, Thoracic and Vascular Surgery, University Hospital of Leipzig AöR, 04103 Leipzig, Germany; rene.thieme@medizin.uni-leipzig.de; 5Department of Pathology, German Medical Institute, 4108 Limassol, Cyprus; andreas.mamilos@goc.com.cy; 6Institute of Pathology, University of Regensburg, 93053 Regensburg, Germany; 7Faculty of Medicine, European University Cyprus, 1516 Nicosia, Cyprus

**Keywords:** CKAP4, gastrointestinal malignancies, biomarker, molecular therapy

## Abstract

**Simple Summary:**

CKAP4, originally identified as an endoplasmic reticulum protein, has been found to localize on the plasma membrane and function as a receptor for various ligands, notably Dickkopf-1 (DKK1). The interaction between DKK1 and CKAP4 activates the PI3K/AKT signaling pathway, promoting cancer cell proliferation and metastasis in various GI malignancies, including esophageal, gastric, pancreatic, and colorectal cancers. The review highlights new findings on CKAP4′s involvement in tumor progression, its potential as a diagnostic biomarker, and its promise as a therapeutic target. These findings could influence future studies by directing research towards developing CKAP4-targeted therapies, exploring its potential as a serum biomarker for early cancer detection, and investigating combination therapies that include CKAP4 inhibition.

**Abstract:**

Cytoskeleton-associated protein 4 (CKAP4) has emerged as a critical player in gastrointestinal (GI) cancer progression, diagnosis, and therapy. This comprehensive review synthesizes current knowledge on CKAP4′s multifaceted roles across GI malignancies, providing novel insights into its mechanisms of action and clinical potential. Its interaction with DKK1 and subsequent activation of the PI3K/AKT pathway underscores its role in promoting tumor growth. This review also highlights novel insights into CKAP4′s mechanisms of action beyond the well-established DKK1-CKAP4 axis, including its interaction with integrin β1 and involvement in angiogenesis through the FMNL2/EGFL6/CKAP4/ERK pathway. CKAP4′s impact on tumor microenvironment and immune evasion is elucidated, offering a new perspective on its contribution to cancer progression. In addition, CKAP4 arises as a promising serum biomarker for early detection and prognosis across multiple GI cancers, emphasizing its potential superiority over traditional markers. The therapeutic potential of targeting CKAP4 is extensively explored, including novel approaches like anti-CKAP4 antibodies and aptamers, and their synergistic effects with existing treatments. By integrating findings from esophageal, gastric, pancreatic, and colorectal cancers, this review provides a unique, comprehensive overview of CKAP4 in GI oncology, underscoring CKAP4′s potential to revolutionize GI cancer diagnosis and treatment and paving the way for future translational research.

## 1. Introduction

Cytoskeleton-associated protein 4 (CKAP4), also named CLIMP-63 or ERGIC-61, is a reversibly palmitoylated, non-glycosylated type II transmembrane protein that was first described as a protein located in the rough endoplasmic reticulum (ER) [1,2]. CKAP4 promotes the formation of clusters in rough ER, while its ability to bind to microtubules makes CKAP4 crucial for the stabilization of the ER [1]. CKAP4 is post-translationally phosphorylated by protein kinase C and casein kinase II [3] and its phosphorylation is crucial for its ability to bind to microtubules [4,5] (Figure 1A).

CKAP4 is also located at the plasma membrane (PM), acting as a cell surface membrane protein (1, 2, 6). Thus, CKAP4 acts as a receptor for various proteins, amongst them tissue plasminogen activator (tPA), surfactant protein A (SP-A), antiproliferative factor (APF), and Dickkopfs’ (DKKs) proteins (1, 2, 7). The reversible palmitoylation of CKAP4 plays a crucial role in its functions, contributing to its transportation to the lipid rafts of the PM (2, 6) (Figure 1B).

The various ligands of CKAP4 reflect its role in multiple signaling pathways that regulate physiological biologic processes such as cell proliferation and migration. The normal function of CKAP4 has been associated with various non-cancerous conditions, including drug-induced cytotoxicity, interstitial cystitis/painful bladder syndrome, as well as atherosclerosis and structural heart diseases [1,6].

Notably, CKAP4 interaction with oncogenic ligands, such as DKK1, regulates signaling pathways involved in cancer. The interaction between DKK1 and CKAP4 activates the PI3K/AKT signaling pathway, promoting cancer cell proliferation [1,2,7,8]. It seems that CKAP4 may have an emerging role in various malignancies such as bladder cancer, and cervical carcinoma, colorectal cancer, lung, pancreatic, esophageal, hepatic, and renal tumors [1,6]. It is reported that its action may either oncogenic by promoting tumor progression or tumor suppressing [6] (Table 1 and Table 2).

Furthermore, CKAP4 has been detected in rat as well as human serum in various cancers like hepatocellular carcinoma (HCC), lung cancer, and pancreatic cancer [13,22,33,35] (Table 3).

Indeed, CKAP4 located in the plasma membrane has been found to be secreted via exosomes by pancreatic duct adenocarcinoma (PDAC) and lung cancer cells both in vitro and in vivo, resulting in detection of CKAP4 in mice and human serum [13,31]. Furthermore, the palmitoylation of CKAP4 increases its secretion from plasma membrane via exosomes [31] (Figure 1C).

The aim of this manuscript is to summarize the current knowledge of CKAP4′s role in several gastrointestinal (GI) malignancies, emphasizing its potential role as a diagnostic biomarker and future therapeutic target.

## 2. CKAP4′s Role in GI Cancer Pathogenesis, Diagnosis, and Prognosis

### 2.1. CKAP4 in Esophageal Cancer

It has been reported that the expression of CKAP4 is elevated in esophageal squamous cell carcinoma (ESCC) specimens and it is correlated to poorer prognosis [7]. The synchronous presence of CKAP4 and DKK1 in ESCC specimens resulted in even worse prognosis. Furthermore, CKAP4′s interaction with DKK1 is shown to play a significant role in ESCC cell proliferation and tumor formation via protein kinase B (Akt) phosphorylation, while the presence of both molecules is the necessary condition for their mechanism of action [7]. Similar results were reported regarding different DKK proteins, like DKK3 [9]. CKAP4′s high expression in ESCC together with activation of the axis DKK1-CKAP4 or DKK3-CKAP4 leading to downstream Akt-activation enhances ESCC cell proliferation and tumor progression, resulting in poorer overall prognosis [9]. Serum CKAP4 levels have also been found to be higher in ESCC patients than healthy ones. In addition, higher serum CKAP4 levels are linked to worse pathologic stage (pT, pN) and reduced DFS [36].

Hence, DKK1-CKAP4 signaling activation promotes tumor growth in ESCC [7,9,10]. Excess amounts of DKK1 are binding to CKAP4, inducing Akt phosphorylation, which in combination with the MEK-ERK pathway results in FOXM1 upregulation. Interestingly, FOXM1 upregulates DKK1 expression and as such enhances further the DKK1-CKAP4 signaling [10]. These findings signify that the DKK1–CKAP4 axis provides an autonomous mechanism of uncontrolled tumor growth.

Such robust findings on potential activation of DKK1-CKPA4 signaling pathway in esophageal adenocarcinoma (EAC) are still lacking. However, Lyros et al. have pointed out the potential role of CKAP4 receptors to justify the reported DKK1-mediated tumor progression in EAC cell lines [39]. Data on the correlation of DKK1 and CKPA4 tissue expressions with EAC patients′ survival are missing.

### 2.2. CKAP4 in Gastric Cancer

Helicobacter pylori-induced activator protein-1 (AP-1) has been shown to promote gastric tumorigenesis through the activation of DKK1/CKAP4 signaling, leading to PI3K/AKT/mTOR pathway activation [11]. High levels of CKAP4 in gastric cancer tissues and gastric cancer cell cultures correlated to worse overall survival and tumor growth, respectively. The activation of the DKK1–CKAP4 axis was found to be responsible for gastric cancer cell proliferation, colony formation, migration, and invasion via activation of the PI3K/AKT/mTOR downstream signaling [11]. This mechanism signifies CKAP4′s role in mediating the effects of H. pylori infection in cancer progression, implicating that CKAP4 may also be involved in various precancerous conditions. Nonetheless, despite the fact that CKAP4 is involved in several inflammatory conditions such as interstitial cystitis, acute kidney injury, and chronic kidney disease [1,40], no such correlation has been proved yet for precancerous lesions like atrophic gastritis or intestinal metaplasia.

CKAP4 is also increased in the serum of gastric cancer patients and it is reduced after surgery for gastric cancer. Its levels could be used for identification of gastric cancer patients with high sensitivity and specificity. Serum CKAP4 levels are associated with the pathological stage and histological grade of gastric cancer [37].

The DKK1–CKAP4 axis has also been reported to cause immune suppression in the tumor microenvironment in gastric cancer. Activation of the DKK1-CKAP4-PI3K-AKT pathway results in macrophages suppression and, consequently, impedes the anti-tumor activity of cytotoxic CD8+ T-cells and natural killer (NK) cells [12]. On the other hand, the knockdown of the DKK1–CKAP4 axis induces gastric cancer regression in vivo, while it seems to enhance the effectiveness of anti-PD-1 treatments [12].

### 2.3. CKAP4 in Pancreatic Cancer

Pancreatic ductal adenocarcinoma (PDAC) cells have been reported to express and secrete CKAP4, with its function associated with tumorigenesis and induction of proliferation and metastasis through activation of the DKK1–CKAP4 axis [8,13]. The expression of CKAP4 protein has been found elevated in PDAC cells [14]. The synchronous expression of both DKK1 and CKAP4 proteins has been linked to decreased 5-year survival and progression-free survival (PFS) [8].

Similarly to other malignancies, activation of the DKK1–CKAP4 axis induces downstream Akt activation and, along with the MEK-ERK pathway, activation results in FOXM1 upregulation promoting PDAC carcinogenesis [10]. Furthermore, CKAP4 seems to enhance cancer cell migration by binding to β1 integrin and negatively regulating α5β1 integrin’s recycling. In this way, the interaction of α5integin with fibronectin is also impeded, resulting in decreased cell adhesion. Interestingly, this CKAP4 action is propagated to be independent of DKK1 [15].

In PDAC, CKAP4 is secreted via exosomes, making it detectable in patient’s sera too. CKAP4 is elevated in serum of patients with PDAC, especially those with elevated CKAP4 in PDAC tissues. In addition, serum CKAP4 levels were higher in the preoperative period and reduced after surgery, while they were significantly higher in unresectable cases [13]. CKAP4 can also serve as a tissue biomarker as it is elevated in PDAC tissue specimens and its expression is correlated to worse prognosis [8,10,13]. This finding suggests that serum CKAP4 could serve as a non-invasive biomarker for both PDAC diagnosis and prognosis.

### 2.4. CKAP4 in Colorectal Cancer

In colorectal cancer (CRC), CKAP4 seems to mediate proliferation and migration as well as induce angiogenesis. Indeed, formin-like 2 (FMNL2), an actin-nucleating protein, interacts with epidermal growth factor-like protein 6 (EGFL6) in order to enhance tumorigenesis, angiogenesis, and metastasis in CRC [16]. CKAP4 plays a critical role in EGFL6′s action, promoting ERK1/2 upregulation and phosphorylation, resulting in increased expression of angiogenic growth factors like VEGFA, VEGFR2, MMP2, and MMP9 [16]. Thus, CKAP4 induces CRC tumorigenesis and migration by promoting angiogenesis through the FMNL2/EGFL6/CKAP4/ERK axis [16].

On the other hand, integrin alpha 7 (ITGA7) is known to be a CRC suppressor. ITGA7 inhibits CRC cell proliferation and migration by binding to CKAP4 and as a result impeding the activation of the PI3K/AKT/NF-κB pathway [17]. Furthermore, DKK1–CKAP4 axis activation seems to be pivotal in CRC resistance in oxaliplatin. The DKK1–CKAP4 interaction induces Akt signaling, resulting in oxaliplatin resistance in vitro and in vivo. At the same time, knockdown of the DKK1/CKAP4/Akt pathway rescues oxaliplatin-induced cytotoxicity [18].

CKAP4 has also been identified as a receptor for Adipocyte enhancer-binding protein 1 (AEBP1) on cancer-associated fibroblasts (CAFs). AEBP1 is predominantly expressed in CAFs within the tumor microenvironment and is positively correlated with PD-L1 expression, T-cell dysfunction, and poor patient survival. Mechanistically, AEBP1 binds to CKAP4 on CAFs, activating the PI3K/Akt signaling pathway, leading to increased PD-L1 expression and T-cell dysfunction [41].

Serum CKAP4 was detected in CRC patients as well. Serum levels of CKAP4 are elevated in CRC patients and reduced after curative surgery, possibly showing great diagnostic effectiveness compared with the usual serum diagnostic markers for CRC. At the same time, higher serum CKAP4 is correlated with worse pathological stage, histological grade, and bigger tumor diameter [38].

These findings suggests that serum CKAP4 could serve as a non-invasive biomarker for both GI cancer diagnosis and prognosis, while its expression in human specimen could also offer a prognostic value.

### 2.5. CKAP4 in Liver Malignancies

Despite the clear oncogenic role of CKAP4 in other GI malignancies, its role in liver malignancies remains contentious due to inconsistent findings across various studies. In the context of HCC, CKAP4 expression has been significantly elevated in HCC tumor tissues compared to adjacent normal liver tissues. Additionally, CKAP4 is upregulated in HCC cell lines and in the sera of HCC patients [19,20,22,23,24,25,35,42]. High expression of CKAP4 is correlated with several clinicopathological features indicative of tumor progression, including elevated α-fetoprotein (AFP) levels, increased inflammation in adjacent liver tissue, poorer tumor histological grades, higher Ishak fibrosis scores, more advanced tumor-node-metastasis (TNM) stage, intrahepatic metastases, and portal venous invasion [20,25].

However, evidence also suggests that high CKAP4 expression correlates with a favorable overall survival (OS) and longer disease-free survival (DFS) for HCC patients [25]. Moreover, patients exhibiting high expression of both CKAP4 and its palmitoyl acyltransferase DHHC2 demonstrated the best OS and DFS rates, despite the low expression of DHHC2 in HCC tissues. This apparent contradiction between high CKAP4 expression and favorable prognostic outcomes may be attributed to the low expression of DHHC2 in HCC, which is necessary for the proper trafficking and function of CKAP4 [25]. Overexpression of CKAP4 has also been shown to inhibit cell proliferation, colony formation, and tumor growth in xenograft models, whereas CKAP4 knockdown enhanced these processes. The inhibition of epidermal growth factor receptor (EGFR) signaling by CKAP4 appeared to be crucial for these outcomes [24].

In other studies, increased CKAP4 expression has been linked to poor prognosis in HCC patients, resulting in reduced OS and DFS [19,20,23,42]. Notably, one study reported significantly higher CKAP4 expression in patients who succumbed within one year compared to those who survived beyond three years [20]. The DKK1–CKAP4 signaling axis has been identified as a crucial factor in enhancing the aggressiveness of HCC tumors. The interaction between DKK1 and CKAP4 is essential for the proliferation of HCC cells, suggesting that the binding of these molecules is a critical step in tumor progression. This interaction promotes cancer cell proliferation, migration, and invasion by activating the PI3K/AKT signaling pathway. Furthermore, the concurrent expression of DKK1 and CKAP4 in HCC patients is correlated with a poorer prognosis compared to those expressing either marker alone or neither [19].

The DKK1–CKAP4 axis was also found to exert immunomodulatory effects in HCC through the activation of the Akt/β-catenin signaling pathway. Indeed, activation of the DKK1–CKAP4 axis is positively correlated with PD-L1 expression and inversely correlated with the infiltration of CD8+ T-cells in the tumor microenvironment. Furthermore, Akt-dependent β-catenin phosphorylation is essential for PD-L1 expression [21].

CKAP4, in addition to serving as a structural component of the ER, appears to play a crucial role in regulating ER-selective autophagy, also known as reticulophagy. CKAP4 directly interacts with RETREG1 (Reticulophagy Regulator 1) within the ER, thereby preventing proteasomal degradation by reducing K48-linked polyubiquitination. Conversely, TRIM21 targets RETREG1 for degradation, thereby inhibiting reticulophagy. CKAP4 and TRIM21 compete for binding to RETREG1, thereby modulating reticulophagy. CKAP4 knockdown impedes the formation and progression of HCC, underlining its significant oncogenic role. Such pro-tumorigenic effects of CKAP4 are also attributed to its capacity to stabilize RETREG1 [42].

CKAP4 is significantly elevated in the serum of patients with HCC [22,35,42]. Furthermore, its levels increase progressively throughout the stages of HCC carcinogenesis, beginning with hepatitis and advancing to cirrhosis and HCC [35]. Notably, CKAP4 may exhibit greater sensitivity than alpha-fetoprotein (AFP) in diagnosing HCC [35], and the combination of CKAP4 and AFP could enhance both sensitivity and accuracy [22]. Therefore, serum CKAP4 presents promising potential for the early diagnosis of HCC.

Controversial findings have been observed in cholangiocarcinoma (CCA) too. Plasmalemma vesicle-associated protein (PLVAP) facilitates angiogenesis in CCA via the DKK1/CKAP4/PI3K/PLVAP signaling pathway [26]. Concurrently, although CKAP4 is overexpressed in intrahepatic CCA, patients exhibiting low CKAP4 expression experience reduced OS and increased recurrence rates. Furthermore, CKAP4 expression is diminished in lymph node metastases compared to primary tumors, indicating that CKAP4 may have a role in the metastatic progression of CCA [27].

This discrepancy underscores the complex and potentially context-dependent role of CKAP4 in cancer progression and highlights the necessity for further research to elucidate its precise function in GI cancer, and especially in liver cancer.

## 3. Targeting CKAP4 in GI Malignancies as a Therapeutic Alternative

Due to the oncogenic function of CKAP4 in GI malignancies, the development of targeted therapies against CKAP4 could offer a new hallmark in treatment of GI cancer. Knockdown of CKAP4 inhibited cell proliferation and suppressed Akt activity in vitro in PDAC and ESCC [7,8,9,10], while it also impeded tumor formation in vivo [8,9]. CKAP4′s knockdown also suppressed migration of PDAC cells and enlarged cell adhesions more efficiently than DKK1′s knockdown [15].

In CRC, sh-RNAs targeting CKAP4 have the ability to suppress both DKK1 and Akt levels, and tumor and colony formation [18]. In addition, knockdown of CKAP4 restrains CRC cell migration and decreases the expression of proliferative factors like p-ERK, VEGFA, and VEGFR2 [16].

The development of humanized anti-CKAP4 antibodies has shown effectiveness in suppressing tumor formation in murine models, offering a new therapeutic avenue. The anti-CKAP4 antibodies impede DKK1 binding to CKAP4, inhibiting Akt phosphorylation and PDAC cell proliferation in vitro and in murine models [8,13,14]. However, these anti-tumor results were evident in cancer cells expressing both DKK1 and CKAP4 and not only CKAP4, or neither DKK1 nor CKAP4 [13,14]. In murine models, the anti-CKAP4 antibodies repressed lymph node metastasis of pancreatic cancer and increased survival [13].

Moreover, anti-CKAP4 antibodies inhibited HCC growth and reduced HCC tumor size and weight [19,42], while their anti-tumor effects were significantly enhanced when combined with lenvatinib, a multikinase inhibitor commonly used in HCC treatment [19]. This suggests that targeting the DKK1–CKAP4 axis could be a promising therapeutic strategy for treating advanced HCC. Humanized anti-CKAP4 antibodies can also inhibit other CKAP4 ligands involved in HCC, like RETREG1 [42].

An anti-CKAP4 antibody has been studied in ESCC as well. Its use in ESCC cells consequently caused the inhibition of Akt activation and cancer cell formation and proliferation. Nonetheless, in cancer cells expressing CKAP4 but not DKK1 the anti-CKAP4 antibody did not exhibit such efficiency [7]. Moreover, an anti-CKAP4 polyclonal antibody blocked DKK3-CKAP4 binding, inhibiting tumor growth [9].

The anti-CKAP4 antibodies amplify PDAC tumor growth inhibition when used in combination with standard chemotherapy regimens, such as gemcitabine and nab-paclitaxel, in vitro and in murine models [14]. In addition, targeting CKAP4 can reduce Akt activation and cancer proliferation in gemcitabine-resistant PDAC cells [13]. Similarly, in CRC cells, DKK1-CKAP4 signal is critical for developing resistance to oxaliplatin. Knockdown of either DKK1 or CKAP4 results in impaired Akt activity and colon cancer formation in cancer cells resistant to oxaliplatin. A specific protein that interferes in the binding of DKK1 to CKAP4 attenuates oxaliplatin-resistant CRC growth [18].

Apart from this, the use of antibodies against CKAP4 results in immunomodulatory effects. They can increase the infiltration of CD8+ cells (cytotoxic T-cells) in the tumor microenvironment, improving anti-tumor immune reaction and suppressing tumor progression [14]. In gastric cancer, the DKK1/CKPA4/Akt axis causes immune suppression in macrophages. Thus, targeting this molecular pathway activates cytotoxic CD8+ T and natural killer (NK) cells, inducing tumor cell apoptosis [12]. Moreover, a small molecule inhibitor which disrupts the interaction between AEBP1 and CKAP4 has been studied in colorectal cancer cells and mouse models. This compound effectively enhances anti-tumor immunity and its combination with immune checkpoint blockade showed superior efficacy compared to either treatment alone [41]. Knockdown of CKAP4 in HCC cells impeded DKK1-mediated Akt activation, decreasing PD-L1 expression [21]. Targeting CKAP4 in combination with PD-1 blockade may offer a novel therapeutic option, which could ameliorate the efficacy of anti-PD-1 regimens.

Targeting CKAP4 in gastrointestinal cancers appears promising given its involvement in cell migration and the expression of proliferative factors, which suggests a variety of mechanisms of action. CKAP4 is implicated in the regulation of several signaling pathways associated with tumorigenesis, including the DKK1/CKAP4/Akt axis, ERK pathway, and VEGF expression. Moreover, the potential efficacy of anti-CKAP4 treatments, particularly in combination with existing therapies, their capacity to overcome drug resistance in certain cancer types, and their possible immunomodulatory effects, could offer significant advantages in clinical settings.

Nevertheless, despite these encouraging preclinical findings, careful consideration is required to ensure the safe and effective clinical application of CKAP4-targeted therapies, as their clinical feasibility has not yet been fully explored. In murine models, knockout mice did not exhibit any abnormal phenotypes, damage to major organs, disrupted blood tests, or abnormal pathological histology [13,14,18,41]. Their body weight was not affected too [42]. Additionally, CKAP4 may represent a more suitable target than DKK1 within the DKK1–CKAP4 axis, due to DKK1′s critical roles in maintaining bone and intestinal tissue homeostasis [7].

However, these assessments are preliminary and do not comprehensively address the safety concerns associated with targeting CKAP4. Further research is essential to develop specific and safe treatments. The development of small molecule inhibitors targeting CKAP4 interactions and the identification of patient subgroups most likely to benefit from CKAP4-targeted therapies could enhance the potential clinical significance of anti-CKAP4 treatments in gastrointestinal cancer.

## 4. Discussion

Cytoskeleton-associated protein 4 (CKAP4) has garnered significant attention in cancer research due to its significant role in tumor progression, diagnosis, and therapy. Originally identified as an ER protein, CKAP4 has been found to localize on the plasma membrane, functioning as a receptor for various ligands, notably DKK1 [2].

The interaction between DKK1 and CKAP4 activates the PI3K/AKT signaling pathway, promoting cancer cell proliferation. This mechanism has been observed in pancreatic and lung cancers, where co-expression of DKK1 and CKAP4 correlates with poor prognosis and reduced relapse-free survival [8]. The DKK–CKAP4 axis seems crucial for CKAP4′s oncogenic role in ESCC, gastric cancer, and CRC [7,9,10,11,12,17,18]. The interaction between DKK1 and CKAP4 promotes metastasis by enhancing cancer cell migration and evasion [2].

In PDAC cells, the cell-surface CKAP4 seems to regulate cell adhesions and migration by controlling β1-integrin recycling independent of DKK1 [15]. Nonetheless, CKAP4 is also an ER protein with a critical role in the cluster formation of ER sheets, keeping them close to the nucleus in a proper, parallel, or antiparallel manner [2]. Indeed, the process of cell migration depends on the structural polarity of the cell, which is controlled by the steepness of the ER-PM contact gradient [43]. CKAP4 regulates the organization and composition of the contact sites of ER and PM, and as a result affects cell migration [43]. In bladder cancer, CKAP4 promotes cancer cells mobility, migration, and tumor metastasis by organizing the cell surface stiffness in an increasing gradient from center to periphery [29]. Such findings in GI malignancies have not yet been established, but further studies in this field may reveal another interesting role of CKAP4 in carcinogenesis of GI cancer.

Moreover, CKAP4 forms condensate in response to solid stress. These condensates directly interact with microtubules, facilitating their branching and reorganization. Consequently, this interaction enhances cancer cell motility, thereby promoting metastasis [44]. CKAP4 is a key regulator of reticulophagy, while TRIM21, its competitor for binding to RETREG1, is upregulated under ER stress [42].

Thus, DKK-CKAP4 is not the only molecular pathway involved in CKAP4-induced tumorigenesis in GI malignancies. In CRC, CKAP4 is also involved in angiogenesis enhancing tumor metastasis both in vitro and in vivo. FMNL2 plays a significant role in carcinogenesis and CKAP4 participates in the downstream molecular pathway interacting with EGFL6 so as to activate ERK/MMP pathway [16]. The DKK1/CKAP4/PI3K/PLVAP signaling pathway emerges to enhance angiogenesis in CCA too [26].

Simultaneously, the inhibition of EGFR signaling by CKAP4 results in the suppression of tumorigenesis in HCC [24]. The expression of its palmitoyl acyltransferase, DHHC2, is essential for this function, underscoring the significance of CKAP4′s diverse localizations to its role [25]. Such findings underscore the complex and potentially context-dependent role of CKAP4 in cancer progression and highlight the necessity for further research to elucidate its precise function in GI cancer.

The discovery of new targeted therapies in GI malignancies is an urgent need. Targeting a molecule like CKAP4, which is involved in cancer progression, could offer effective alternative therapeutic options. Indeed, anti-CKAP4 monoclonal antibodies have shown promising results in inhibiting tumor growth, indicating their potential utility in targeted therapy [13]. Anti-CKAP4 antibodies block DKK1 binding, suppressing AKT activity and inhibiting tumor growth in xenograft models [8].

Another hallmark in GI malignancies treatment could be the combination of CKAP4-targeted therapies with standard treatments such as chemotherapy or immunotherapy. This could improve outcomes and reduce treatment-resistance in GI cancers. The anti-CKAP4 antibodies have been studied in combination therapy with chemotherapy agents with promising results [14]. In addition, targeting the DKK1–CKAP4 axis attenuates cancer growth in gemcitabine-resistant PDAC cells and oxaliplatin-resistant CRC cells [13,18].

Furthermore, the DKK1–CKAP4 axis demonstrates immunomodulatory effects affecting the tumor microenvironment. The activation of the DKK1-CKAP4 pathway leads to restraint of the function of cytotoxic T-cells, NKs, M1 and M2 macrophages, and tumoral myeloid cells, having as consequence a suppressive tumor immune microenvironment [12]. Targeting DKK1 or CKAP4 results in increased accumulation of cytotoxic T-cells in tumor lesions as well as improved ability of the macrophages to activate cytotoxic lymphocytes [12,14]. In hepatocellular carcinoma, DKK1′s interaction with CKAP4 increases PD-L1 expression as well [21]. In fact, in breast cancer, DKK1 expressed by cancer-associated fibroblasts and bone cells suppressed NK cells’ anti-cancerous activity [45]. Similar results could be possibly applied to CKAP4. Thus, the combination of anti-DKK1/CKAP4 and anti-PD-1 treatment may have promising results in anti-cancer treatment [12].

Aptamers are single-stranded oligonucleotides, for example, ssDNA, si-RNA, and mi-RNA, which bind specifically and strongly with molecular targets such as proteins. Aptamer offers a novel and innovative alternative in targeted cancer therapy [46]. Si-RNAs attenuated ESCC cell proliferation in vitro as well as tumor formation in vivo [9]. CKAP4-targeted aptamers successfully inhibit metastasis of bladder cancer cells in preclinical models. CKAP4-targeted aptamers present dual-action blocking of CKAP’s interaction with DKK1 and, thus, disrupting PIK/Akt signaling as well as impeding cancer cell adhesion and migration by decreasing internalization and recycling of α5β1 integrin [46]. Moreover, si-RNAs have shown efficacy in the blockage of DHHC2 and, thus, CKAP4′s translocation to the PM is impeded due to the inhibition of its palmitoylation. The potential use of anti-CKAP4 aptamers could launch a new era in GI malignancies therapy (Figure 2).

Furthermore, CKAP4 can also be found in animal and human serum. Serum CKAP4 has been proven to be elevated in patients with PDAC, ESCC, gastric cancer, and CRC. Its levels were also correlated to pathologic stage and prognosis [33,37,38]. Release from cancer cells via exosomes as well as tumor expression levels play a significant role in the presence of CKAP4 in the serum of cancer patients [13,31]. Interestingly, similar cut-off values (0,25–0,75 ng/mL) were proposed in different cancer types [22,35,36,37,38]. In addition, serum CKAP4 had comparable or even better sensitivity and specificity than traditional biomarkers in GI and liver malignancies like AFP, CEA, and CA19-9 [22,38]. The integration of CKAP4 with additional biomarkers may further enhance sensitivity, specificity, and prognostic value [23,35]. These findings suggest that serum CKAP4 could serve successfully as a non-invasive biomarker for GI cancer diagnosis and follow-up.

Further studies in animal models, human specimens, as well as clinical trials are required so that the potential of CKAP4 as a biomarker and therapeutic target is further explored. According to the data presented in this review, this field is really promising.

## 5. Conclusions

CKAP4′s role is pivotal in the carcinogenesis of GI malignancies promoting cancer proliferation through various mechanistic pathways. Several preclinical and clinical studies underline its promising potential as a diagnostic biomarker and therapeutic target. Further research is warranted to fully elucidate CKAP4′s functions and to translate these findings into effective clinical applications.

## Figures and Tables

**Figure 1 curroncol-32-00561-f001:**
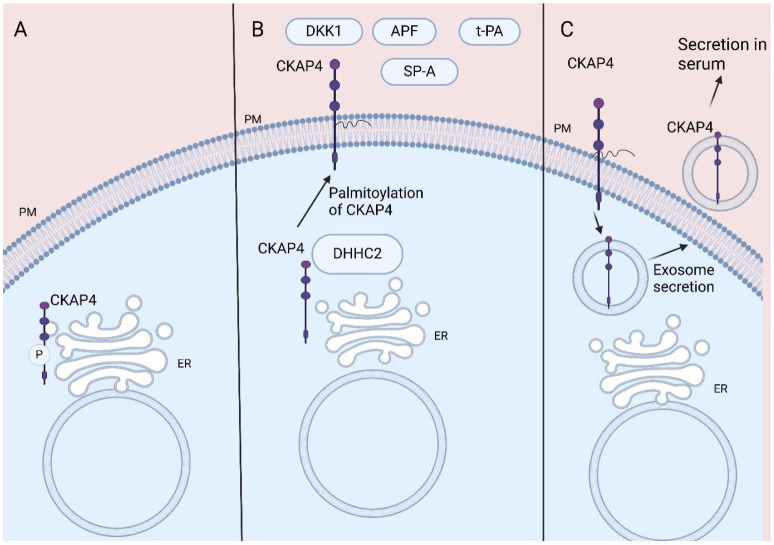
CKAP4 location: (**A**) CKAP4 binds to microtubules stabilizing ER. CKAP4′s phosphorylation is crucial for its role. (**B**) CKAP4′s palmitoylation leads to its transportation to PM acting as receptor for various ligands. (**C**) CKAP4 located in PM is secreted via exosomes, resulting in its detection in serum. CKAP4, Cytoskeleton-associated protein 4; ER, endoplasmic reticulum; PM, plasma membrane; DKK1, Dickkopf 1; APF, antiproliferative factor; t-PA, tissue plasminogen activator; SP-A, surfactant protein A. Created in BioRender 201. Despotidis, M. (2025) https://BioRender.com/mhuvdvf (accessed on 2 September 2025).

**Figure 2 curroncol-32-00561-f002:**
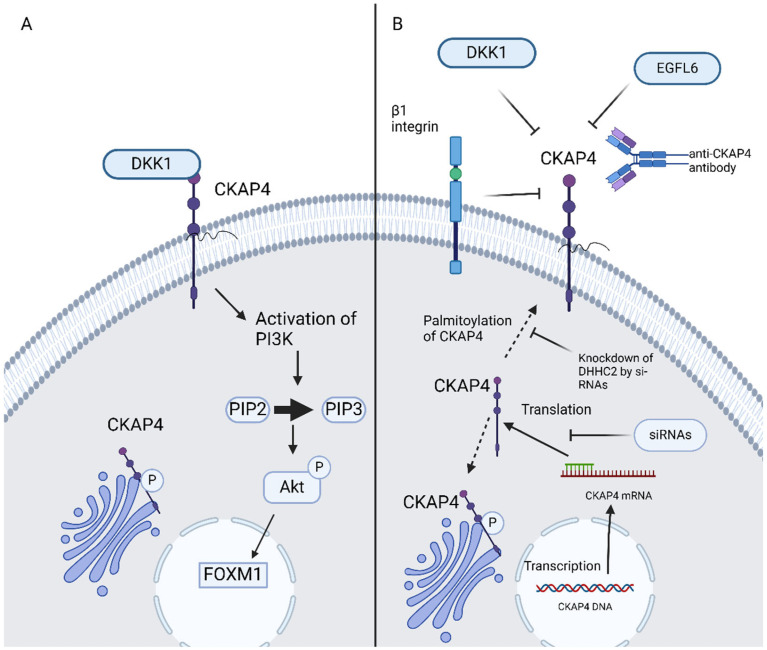
(**A**) DKK1–CKAP4 axis, (**B**) possible ways of inhibition of CKAP4 signaling in GI malignancies. CKAP4, cytoskeleton-associated protein 4; DKK, Dickkopf; EGFL6, epidermal growth factor-like protein 6; PIP2, phosphatidylinositol (4,5)-bisphosphate; PIP3, phosphatidylinositol (3,4,5)-trisphosphate; AKT, protein kinase B; FOXM1, forkhead box M1; siRNAs, small interfering RNAs. Created in BioRender 201. Despotidis, M. (2025) https://BioRender.com/e5q6awl (accessed on 2 September 2025).

**Table 1 curroncol-32-00561-t001:** Role of CKAP4 in GI cancer.

Type of Tumor	CKAP4 Role	Downstream Mechanism	Studies
Esophageal cancer	Oncogene [7,9,10]	DKK -CKAP4 axis [7,9,10]	Shinno et al. [7], Kajiwara et al. [9], and Kimura et al. [10].
Gastric cancer	Oncogene [11,12]	DKK -CKAP4 axis [11,12]	Luo et al. [11] and Shi et al. [12].
Pancreatic cancer	Oncogene [8,10,13,14,15]	DKK -CKAP4 axis [8,10,13,14] CKAP4-β1 integrin, independent of DKK1 [15]	Kimura et al. [8], Kimura et al. [10], Kimura et al. [13], Sada et al. [14], and Osugi et al. [15].
Colorectal cancer	Oncogene [16,17,18]	FMNL2/EGFL6/CKAP4/ ERK axis [16] DKK -CKAP4 axis [17,18]	He et al. [16], Wang et al. [17], and Hsieh et al. [18].
Hepatocellular carcinoma	Oncogene [19,20,21,22,23] Tumor suppressor [24,25]	DKK-CKAP4 axis [19] Upregulation of PD-L1 [21] Suppression of EGFR signal [24] CKAP4 palmitoylation by DHHC2 [25]	Iguchi et al. [19], Chen et al. [20], Yang et al. [21], and Wang et al. [22]. Li et al. [24] and Li et al. [25].
Cholangiocarcinoma	Oncogene [26] Tumor suppressor [27]	DKK-CKAP4 axis [26] Suppression of EGF signal [27]	Wang et al. [26]. Li et al. [27].

**Table 2 curroncol-32-00561-t002:** Role of CKAP4 in non-GI cancer.

Type of Tumor	CKAP4 Role	Downstream Mechanism	Studies
Renal cancer	Oncogene [28]	CKAP4-Cyclin B Signal [28]	Sun et al. [28]
Bladder cancer	Oncogene [29] Tumor suppressor [30]	Regulating stiffness on the cell membrane [29] APF-CKAP4 signal [30]	Sun et al. [29] Shahjee et al. [30]
Lung cancer	Oncogene [8,31,32,33]	DKK-CKAP4 axis [8,31] GOLPH3-CKAP4 signal [32]	Kimura et al. [8], Nagoya et al. [31], Song et al. [32], and Yanagita et al. [33]
Glioma	Tumor suppressor [34]	Targeting the miR-671-3p [34]	Lu et al. [34]

**Table 3 curroncol-32-00561-t003:** Serum CKAP4 and cancer prognosis.

Type of Tumor	Serum CKAP4	Prognostic Value	Studies
Lung cancer	Elevated in mice [31] and human patients [31,33]	Correlated to worse PFS and OS [31] and to distant metastasis [33]	Nagoya et al. [31] and Yanagitael al. [33]
Hepatocellular carcinoma	Elevated in mice [35] and human patients [22,23,35]	Higher expression levels associated with poor prognosis [23]	Li et al. [35] and Wang et al. [22]
Pancreatic adenocarcinoma	Elevated in mice and human patients [13]	Higher in unresectable cases [13]	Kimura et al. [13]
Esophageal squamous cell carcinoma	Elevated in human patients [36]	Correlated to worse PFS [36]	Chen et al. [36]
Gastric cancer	Elevated in human patients [37]	Correlated to worse pathological stage and histological grade [37]	Peksoz et al. [37]
Colorectal cancer	Elevated in human patients [38]	Correlated to worse disease stage, histological grade, and bigger tumor diameter [38]	Disci et al. [38]

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
