# Peer review of "The Emerging Role of CKAP4 in GI Cancer: From Molecular Pathways to Clinical Applications"

_curroncol, 2025, doi:10.3390/curroncol32100561_

Round 1
Reviewer 1 Report
Comments and Suggestions for Authors
This review paper is of good quality. However, there are two major concerns, which should be easily addressed by authors:
1. The lack of inclusion of recent important papers in this field. These are the few that I could identify, but I advise the authors to conduct another literature search for the last two years (specifically 2024 and 2025) just to be sure that none of the important papers are missed. Pleases see:
- PMID: 39528501
- PMID: 39118263
- Cancer Res (2025) 85 (8_Supplement_1): 4273.
- PMID: 40673276
2. The two figures are not original in content because this is a review paper, not original paper. Although this is expected for the review paper, it is not acceptable to miss source acknowledgment (e.g., "adapted from"). This is critical because we do not want authors to be accused of academic misconduct. Similar critical assessment needs to be done for the tables. If there is significant similarity or duplication from the previously published papers, please incorporate the source acknowledgment.
Author Response
Dear Reviewer,
Thank you for your detailed review. To address your concerns:
Comment 1.
The lack of inclusion of recent important papers in this field. These are the few that I could identify, but I advise the authors to conduct another literature search for the last two years (specifically 2024 and 2025) just to be sure that none of the important papers are missed. Pleases see:
- PMID: 39528501
- PMID: 39118263
- Cancer Res (2025) 85 (8_Supplement_1): 4273.
- PMID: 40673276
Response 1.
A comprehensive literature review is essential to understanding the role of CKAP4 in gastrointestinal (GI) malignancies, especially considering its emerging significance. This is why numerous papers published after 2023 have been included. As for the specific papers you mentioned, PMID: 39118263 is already included (Ref. 14), while the others have been carefully reviewed and incorporated into the paper as appropriate.
Comment 2.
The two figures are not original in content because this is a review paper, not original paper. Although this is expected for the review paper, it is not acceptable to miss source acknowledgment (e.g., "adapted from"). This is critical because we do not want authors to be accused of academic misconduct. Similar critical assessment needs to be done for the tables. If there is significant similarity or duplication from the previously published papers, please incorporate the source acknowledgment.
Response 2.
I appreciate your attention to the figures and tables, as proper source acknowledgment is vital in scientific research. In fact, the two figures have been created by us using Biorender and the publication licence has been attached to the submission papers. Regarding the tables, they were also crafted by our team specifically for this paper and they refer to the role of CKAP4 in cancer.
We sincerely thank you for your remarks on our manuscript, "The Emerging Role of CKAP4 in GI Cancer: From Molecular Pathways to Clinical Applications". We appreciate your thoughtful suggestions and believe that implementing them has substantially improved our work. If there are any additional suggestions or clarifications needed, we will be pleased to address them.
Yours sincerely,
Markos Despotidis
On behalf of the Author team
Reviewer 2 Report
Comments and Suggestions for Authors
The manuscript sets out to provide a comprehensive overview of CKAP4 (cytoskeleton-associated protein 4) in gastrointestinal (GI) cancers, spanning its molecular functions to its potential clinical applications. The subject matter is moderately original and timely, as there is increasing interest in identifying novel biomarkers and therapeutic targets for GI malignancies. While the focus on CKAP4 within this context is relatively narrow, it touches upon an underexplored niche that holds translational promise. However, for the manuscript to make a significant contribution to the field, it must offer more than a summary of existing knowledge. Specifically, it needs to generate new insights, ideally GI cancer-specific interpretations or comparative analyses across different tumor types, to distinguish it from prior reviews on CKAP4 in broader cancer biology.
Although the conceptual framework of the manuscript is sound, the current execution lacks the depth, synthesis, and organizational rigor necessary to elevate it beyond a literature compilation. The authors should aim for a more critical, integrative approach that connects basic molecular insights to translational and clinical utility. As it stands, the manuscript risks redundancy with prior reviews unless it enhances both the analytical depth and comprehensiveness of the discussion.
Importantly, CKAP4 is not a newly discovered molecule, and its roles in various cancers have already been documented. Therefore, a fresh perspective—particularly one focused on its implications specifically in GI malignancies—is needed. The review could benefit from synthesizing emerging GI-specific data or reinterpreting known mechanisms through the lens of gastrointestinal tumor biology.
The authors’ emphasis on CKAP4 as a potential biomarker, especially within the context of liquid biopsy, is well-aligned with the current push toward non-invasive diagnostics and precision oncology. Likewise, the discussion of CKAP4 as a therapeutic target, particularly in combination with standard-of-care treatments, is highly relevant in the era of personalized medicine. Notably, recent studies over the past 2–3 years have highlighted CKAP4's role in immune regulation and metastasis, making this an opportune moment to explore its therapeutic implications. However, the manuscript falls short in bridging molecular findings to real-world clinical application, which weakens its translational impact.
In summary, while the manuscript addresses a compelling and underexplored aspect of GI cancer biology, its current form does not fully realize the potential of the topic. To enhance its impact and originality, the authors should adopt a more critical and GI-specific approach, structure their findings more clearly, and engage with the translational implications of CKAP4.
The authors should focus on the following key areas for improvement:
- The abstract lacks focus and specificity. It should clearly articulate the unique contribution of this review, highlight any novel insights or reinterpretations, and differentiate the work from existing literature.
- The tables summarizing CKAP4's role in different cancers need standardization. The inconsistent use of acronyms versus full cancer names hinders readability and clarity. More importantly, given the declared focus on GI malignancies, the content and emphasis of the tables should be restricted accordingly or clearly distinguished from non-GI contexts.
- The absence of a dedicated subsection on hepatocellular carcinoma in Section 2 is a significant omission, particularly since conflicting roles of CKAP4 in HCC are mentioned elsewhere in the manuscript. These contradictory findings should be addressed explicitly in the text, with discussion on potential context-dependent mechanisms or experimental variability.
- Section 3 presents promising preclinical findings from cell line and xenograft models. However, its current focus is narrowly confined to early-stage experimentation. A meaningful review must also evaluate the translational potential of CKAP4-targeted interventions. This includes addressing the feasibility of clinical application, the protein’s context-specific roles, its integration with known signaling pathways, and, most notably, the specificity and safety of CKAP4-targeted therapies considering CKAP4’s physiological roles in normal cells. Without such considerations, the therapeutic discussion lacks clinical relevance and feels disconnected from authors’ goals.
Author Response
Dear reviewer,
We would like to express our gratitude for your valuable feedback on our manuscript, "The Emerging Role of CKAP4 in GI Cancer: From Molecular Pathways to Clinical Applications". We have carefully addressed each of the comments and revised the manuscript accordingly. Below, we provide detailed responses and explain the modifications made.
Comment 1.
The abstract lacks focus and specificity. It should clearly articulate the unique contribution of this review, highlight any novel insights or reinterpretations, and differentiate the work from existing literature.
Response 1.
The abstract is crucial for the impact of the review. The abstract has been modified to sharpen the focus and communicate the specific value of this review to readers.
Comment 2.
The tables summarizing CKAP4's role in different cancers need standardization. The inconsistent use of acronyms versus full cancer names hinders readability and clarity. More importantly, given the declared focus on GI malignancies, the content and emphasis of the tables should be restricted accordingly or clearly distinguished from non-GI contexts.
Response 2.
The acronyms have been replaced by full cancer names and table 1 has been divided in two tables, one including GI cancers and the second inclyding non-GI cancers.
Comment 3.
The absence of a dedicated subsection on hepatocellular carcinoma in Section 2 is a significant omission, particularly since conflicting roles of CKAP4 in HCC are mentioned elsewhere in the manuscript. These contradictory findings should be addressed explicitly in the text, with discussion on potential context-dependent mechanisms or experimental variability.
Response 3.
Aim of our review was mainly to emphasize on CKAP4's oncogenic role in GI tract malignancies and highlight its potential as biomarker and therapeutic target. However, addressing the conflicting role of CKAP4 in liver malignancies is important to fully comprehend the role of CKAP4 in cancer, Therefore this has been clearly addressed in discussion section.
Comment 4.
Section 3 presents promising preclinical findings from cell line and xenograft models. However, its current focus is narrowly confined to early-stage experimentation. A meaningful review must also evaluate the translational potential of CKAP4-targeted interventions. This includes addressing the feasibility of clinical application, the protein’s context-specific roles, its integration with known signaling pathways, and, most notably, the specificity and safety of CKAP4-targeted therapies considering CKAP4’s physiological roles in normal cells. Without such considerations, the therapeutic discussion lacks clinical relevance and feels disconnected from authors’ goals.
Response 4.
Targeted therapies against CKAP4 are primarily explored in preclinical settings. Nonetheless, these considerations are crucial, and as a result section 3 has been enhanced to address them comprehensively.
We sincerely appreciate your insightful review and invaluable feedback, which have significantly enhanced the quality of our manuscript. We are confident that the revisions we have implemented align with your expectations. Should you have any further comments or require additional clarifications, we are fully prepared and eager to address them promptly and thoroughly.
Yours sincerely,
Markos Despotidis
On behalf of the Author team
Round 2
Reviewer 2 Report
Comments and Suggestions for Authors
The authors have addressed most of the concerns raised in a satisfactory manner. However, they have not included a subsection discussing the role of CKAP4 in liver cancer. While they state that their focus is on gastrointestinal tract malignancies, it is worth noting that they have included a subsection on pancreatic cancer, which is not strictly considered part of the GI tract. Moreover, the conflicting findings regarding CKAP4's role in liver cancer further underscore the importance of including this topic in their discussion.
Author Response
Dear reviewer,
Thank you for your feedback on our response.
Comment 1:
The authors have addressed most of the concerns raised in a satisfactory manner. However, they have not included a subsection discussing the role of CKAP4 in liver cancer. While they state that their focus is on gastrointestinal tract malignancies, it is worth noting that they have included a subsection on pancreatic cancer, which is not strictly considered part of the GI tract. Moreover, the conflicting findings regarding CKAP4's role in liver cancer further underscore the importance of including this topic in their discussion.
Response 1:
We appreciate your suggestion regarding the inclusion of a subsection on CKAP4 in liver cancer. A subsection has been incorporated to enhance the comprehensiveness and balance of the review.
Thank you for bringing this to our attention. We remain open to any further suggestions or requests for clarification.